# Hybrid plasmonic nano-emitters with controlled single quantum emitter positioning on the local excitation field

Dandan Ge[1,2], Sylvie Marguet[3], Ali Issa[1], Safi Jradi[1], Tien Hoa Nguyen[4], Mackrine Nahra[1], Jéremie Béal[1], Régis Deturche[1], Hongshi Chen[1,2], Sylvain Blaize[1], Jérôme Plain[1], Céline Fiorini[5], Ludovic Douillard[5], Olivier Soppera[6,7], Xuan Quyen Dinh[4,8,9], Cuong Dang[4,9], Xuyong Yang[10], Tao Xu[10,11✉], Bin Wei[10], Xiao Wei Sun[2,12], Christophe Couteau[1] & Renaud Bachelot[1,2,10,11✉]

Hybrid plasmonic nano-emitters based on the combination of quantum dot emitters (QD) and plasmonic nanoantennas open up new perspectives in the control of light. However, precise positioning of any active medium at the nanoscale constitutes a challenge. Here, we report on the optimal overlap of antenna's near-field and active medium whose spatial distribution is controlled via a plasmon-triggered 2-photon polymerization of a photosensitive formulation containing QDs. Au nanoparticles of various geometries are considered. The response of these hybrid nano-emitters is shown to be highly sensitive to the light polarization. Different light emission states are evidenced by photoluminescence measurements. These states correspond to polarization-sensitive nanoscale overlap between the exciting local field and the active medium distribution. The decrease of the QD concentration within the monomer formulation allows trapping of a single quantum dot in the vicinity of the Au particle. The latter objects show polarization-dependent switching in the single-photon regime.

[1] Light, nanomaterials, nanotechnologies (L2n) Laboratory, CNRS ERL 7004, University of Technology of Troyes, 12 rue Marie Curie, 10004 Troyes, France. [2] Shenzhen Planck Innovation Technologies Co. Ltd, Shenzhen, Guangdong 518116, People's Republic of China. [3] Université Paris Saclay, CEA, CNRS, NIMBE, 91191 Gif sur Yvette, France. [4] CNRS-International-NTU-Thales Research Alliance (CINTRA), 50 Nanyang Drive, Singapore 637553, Singapore. [5] Université Paris Saclay, CEA, CNRS, SPEC, 91191 Gif sur Yvette, France. [6] Université de Haute Alsace, CNRS, IS2M UMR 7361, 68100 Mulhouse, France. [7] Université de Strasbourg, Strasbourg, France. [8] Thales Solutions Asia Pte Ltd, R&T Department, 21 Changi North Rise, Singapore 498788, Singapore. [9] School of Electrical and Electronic Engineering, Nanyang Technological University, Nanyang Avenue, Singapore 639798, Singapore. [10] School of Mechatronic Engineering and Automation, Key Lab of Advanced Display and System Application, Ministry of Education, Shanghai University, Shanghai 2000072, People's Republic of China. [11] Sino-European School of Technology, Shanghai University, Shanghai, People's Republic of China. [12] Department of Electrical and Electronic Engineering, Southern University of Science and Technology, Shenzhen, Guangdong 518055, People's Republic of China. ✉email: xtld@shu.edu.cn; renaud.bachelot@utt.fr

Rapid growth of nanophotonics area requires the development of advanced optical nanosources. Over the past decade, hybrid plasmonic nanosources based on energy transfers between metal nanoparticles and semiconductors quantum dots/nanocrystals or organic dyes have turned out to constitute a promising solution of efficient optical nanosources to be integrated into photonic nanodevices[1–28]. In particular, weak coupling (and associated Purcell effect) between metal nanostructures and quantum systems allows for controlled light emission through the control of the deexcitation rate of the nano-emitters. A physical picture generally used is that the electromagnetic local density of states (LDOS) of the metal nanostructure acts as channels of deexcitation of the nano-emitters, resulting in an increase of the deexcitation rate and a decrease of the lifetime[29–31].

Despite the outstanding achievements that have been reported[1–31], a main challenge remains: to control the nanoscale spatial distribution of nano-emitters relative to the LDOS that can be partially used through the polarization state of the incident field. Taking up this challenge would allow one to use the incident polarization as a fast and efficient remote optical control of light emission from the hybrid emitter. This issue can be discussed on the basis of Eq. (1)

$$\gamma_{\mathrm{em}}(\nu_{\mathrm{em}}) = \gamma_{\mathrm{exc}}(x, y, z, \nu_{\mathrm{exc}}) \times Q(\nu_{\mathrm{em}}) \times \rho(x, y, z)\mathrm{d}V, \qquad (1)$$

where $\gamma_{\mathrm{em}}$ is the rate of emission of the nano-emitter at $\nu_{\mathrm{em}}$ light frequency, $\gamma_{\mathrm{exc}}$ is the rate of excitation. $\nu_{\mathrm{exc}}$ is the frequency of the exciting field that is absorbed by the emitter. In the case of one-photon absorption, $\nu_{\mathrm{exc}}$ should be within the absorption band of the emitter, thus $\nu_{\mathrm{em}} - \nu_{\mathrm{exc}}$ (<0) represents the Stokes shift. $Q$ is the nano-emitter quantum yield and $\rho(x, y, z)\mathrm{d}V$ is the probability of presence of emitters around the metal nanoparticle within an elementary volume $\mathrm{d}V$ (=$\mathrm{d}x\mathrm{d}y\mathrm{d}z$) at position $(x, y, z)$. $\rho(x, y, z)$ is thus the volume density of probability of the nano-emitters presence. It should be pointed out that $\rho(x, y, z)$ can also be considered as the volume density of emitters. Note that $Q$ is related to the probability of light emission once the nano-emitter gets excited, $Q$ varying also with $(x, y, z)$[32]. This dependence will be discussed further. For the moment, let us focus our attention onto the spatial dependencies of $\gamma_{\mathrm{exc}}$ and $\rho$. Emitted light can get amplified by good match with the plasmon resonance of the metal nanoparticle or, in the more practical point of view, by optimizing the geometry of light collection to match it to the far-field radiation diagram of the hybrid nanosource[33,34]. $\gamma_{\mathrm{exc}}$ is related to the metal nanoparticle plasmonic near-field whose spatial distribution can be controlled through incident polarization for a given nanoparticle size and geometry[35–37]. For example, illuminating a small spherical plasmonic particle with an incident linear polarized light leads to an electromagnetic near-field made of two lobes aligned along the incident field polarization[35]. Similarly, an incident polarized light parallel to the long axis of a bowtie plasmonic antenna can lead to the excitation of the gap mode and a resulting hot spot within the gap[37].

The control of the nano-emitters (the active medium in general) spatial distribution $\rho(x, y, z)$ thus constitutes a big challenge. This general concept is not new in micro-optoelectronics[38–40]. For example, Eq. (2) expresses the overlap integral that accesses the way a specific wave guide mode can be excited[38]

$$\eta = \frac{\left|\iint E_1 E_2 \mathrm{d}x\mathrm{d}y\right|^2}{\iint |E_1|^2 \mathrm{d}x\mathrm{d}y \times \iint |E_2|^2 \mathrm{d}x\mathrm{d}y}, \qquad (2)$$

where $\eta$ is the coupling efficiency. $E_1$ and $E_2$ are, respectively, the complex amplitude of the mode to be coupled and the complex amplitude of the incident exciting field. $x$, $y$ are the spatial coordinates within a plane perpendicular to light propagation.

Equation (2) will inspire us to define a new parameter in the context of hybrid plasmonic nano-emitters.

In hybrid plasmonic nanosources, the local optical field spatial distribution cannot be exploited unless the nano-emitter spatial distribution $\rho(x, y, z)$ is also controlled. In most reported cases, the distribution of emitters is isotropic because spin coating is generally used for depositing them onto the plasmonic system[16,41–43]. Spin coating is a quick and simple solution but it does not allow any control of the position of the emitters relative to the metal resonant nanostructure. As a result, many samples must be made before obtaining a satisfactory one where the emitter location is suitable for physical studies of interest[16]. Other studies start from homogeneous nano-emitter deposition followed by subsequent steps in order to fabricate the plasmonic structure around the nano-emitters or to localize them at strategic positions: optical lithography (at the submicron scale)[44], electron beam lithography (at the nanoscale)[45] or atomic force microscopy AFM (at the nanoscale)[46]. A two-step electron beam lithography process combined with chemical functionalization permitted the integration of a single QD at the appropriate site on a plasmonic Yagi Uda antenna[47]. A DNA based approach was also used for attaching nano-emitters[48–50]. Such an approach is powerful, but it does not offer a high flexibility in the nano-emitter positioning in the sense that only gaps between metal nanoparticles can be functionalized.

In general, a simple approach is still needed to obtain, on demand, integrated hybrid plasmonic nanosources with controlled position of the active medium. Recently, we reported a method of plasmon-based photopolymerization to integrate polymer nanostructures containing nano-emitters in the close vicinity of metal nanoparticles[51,52]. It consists of a polymer printing of specific pre-selected local plasmonic modes supported by the metal nanoparticles.

In the present work, we study the spatial overlap between optical near-field and quantum emitter distribution in anisotropic hybrid sources based on Au nanocubes. Three important novelties are stressed. First, we report on controlled infrared 2-photon free radical polymerization on single Au nanocubes. Secondly, we report on nanocube-based hybrid systems whose integrated active medium (containing quantum emitters) is anisotropic, permitting selective excitation of a 2-state system with incident polarization. Nanocube-based hybrid emitters are compared with nanodisk-based systems in terms of polarization sensitivity. Finally, we present a preliminary observation of polarization-sensitive single-QDs hybrid nanosources that were obtained by decreasing the concentration of QDs within the photosensitive formulation.

## Results

**Nano-object fabrication and characterization.** Our plasmonic samples consist of gold nanocubes (single crystal, side length 127 ± 2 nm) deposited on an indium-tin oxide (ITO)-coated glass substrate (see the Methods section for details of the sample preparation). The surfactant used during the synthesis is cetyl-trimethylammonium bromide (CTAB). The nanocubes deposited on a substrate are nano-objects with a $C_{4v}$ in-plane symmetry. A cube possesses three dipolar plasmon eigenstates of $E$ and $A_1$ symmetry, respectively. The two degenerated E modes correspond to the two charge combinations (0, 1, 0, −1, 0, 1, 0, −1) and (1, 0, −1, 0, 1, 0, −1, 0), where the number sequences represent the top and bottom corner positions. These orthogonal in-plane eigenvectors correspond to the diagonals of the top and bottom faces. The $A_1$ dipolar mode corresponds to a unique eigenstate vector. Its polarization is aligned along the $z$ vertical axis and can be described by the charged corner sequence (1, 1, 1, 1, −1, −1, −1, −1). All these resonance states can be selectively

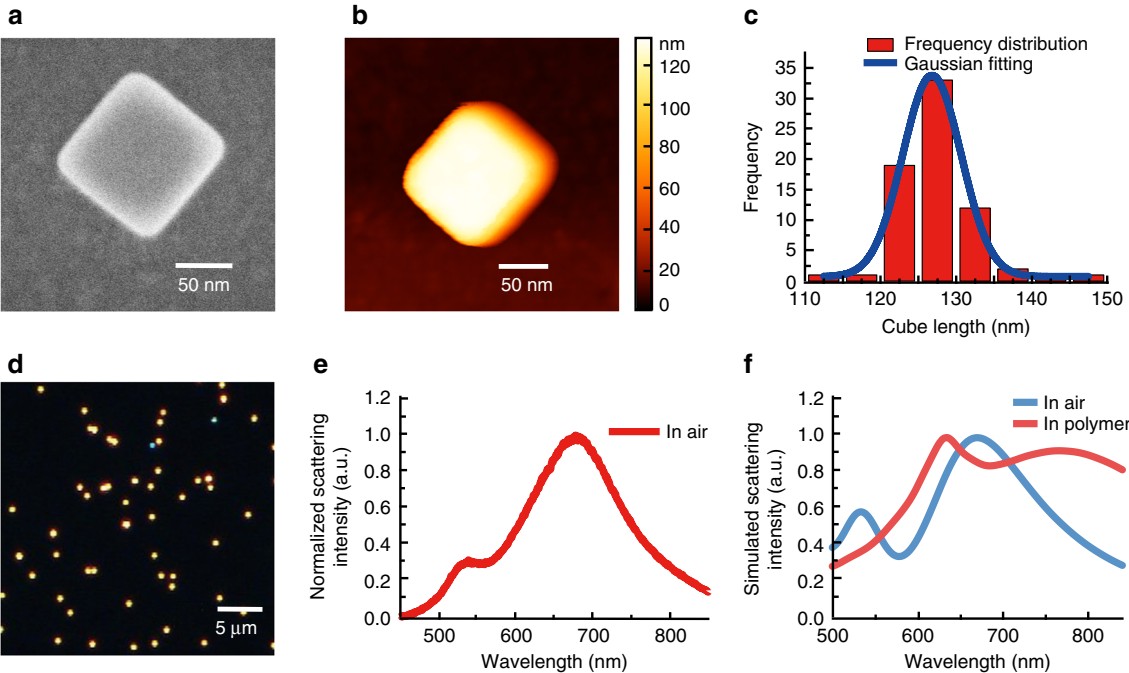

**Fig. 1 Characterization of Au nanocubes. a** SEM image, and **b** AFM image of a representative single Au nanocube (same object). **c** Nanocube size (edge length) histogram obtained from a set of 100 nanocubes (SEM analysis). **d** Dark-field scattering image of Au nanocubes on ITO-coated glass substrate. **e** Dark-field single-nanocube scattering spectrum averaged from 10 nanocubes in air on ITO-coated glass substrate. **f** Calculated scattering spectrum of a single Au nanocube in air and in polymer (refractive index $n = 1.48$) on ITO-coated glass substrate (40 nm thickness of ITO layer with refractive index of 2).

excited via a proper choice of the polarization of the incident light field[53].

In order to characterize the gold nanocubes and the resulting hybrid nano-emitters, scanning electron microscopy (SEM), atomic force microscopy, dark-field white light imaging/spectroscopy and micro-photoluminescence (PL) techniques were used. Appropriate nanocube concentration in solution allowed us to address single nano-objects after deposition and immobilization on ITO-coated glass cover slides (averaged density of 0.1 nanocube μm$^{-2}$). Figure 1 shows a typical set of experimental and numerical characterizations of gold nanocubes presenting an in-plane dipolar plasmonic mode at 680 nm in air (see figure caption and "Methods" section for more details).

Plasmon-assisted 2-photon photopolymerization was conducted on single gold nanocubes with a formulation of 1%wt Irgacure 819 (IRG819 is used as a photo initiator in the 2-photon absorption regime) and 99%wt QDs-grafted pentaerythritol triacrylate (PETA). QDs are red light-emitting CdSe/CdS/Zn core/shell/shell colloidal quantum dots. The QD photoluminescence wavelength is centered at 625 nm. More details of this QD-containing hybrid photosensitive system can be found in references[52,54]. In a typical photopolymerization process, light induces the polymerization reaction when the exposure energy dose exceeds a given threshold $D_{th}$ that is assessed beforehand[55,56]. On the contrary, for plasmonic near-field 2-photon polymerization, an incident exposure dose $D_{in} = p \cdot D_{th}$ ($p < 1$) below the polymerization threshold $D_{th}$ is used. This is to guarantee the selective integration of polymer structures in the close vicinity of the nanostructures. Indeed, upon plasmonic excitation, the highly confined optical near-fields surrounding the particle increases the local effective exposure dose beyond the polymerization threshold. In particular, no far-field polymerization takes place within the formulation drop volume. Details on the process of plasmon-induced 2-photon polymerization on gold nanocubes can be found in the methods section.

Figure 2 shows examples of resulting hybrid structures, obtained with $p = 0.5$. This value of $p$ results from first experimental studies and takes into account the field enhancement factor at the nanocube surface. A parameter study of the influence of $p$ can be found in the Supplementary Note 3. Figure 2a is a SEM image of a representative hybrid nanocube obtained with an incident polarization parallel to the cube diagonal at 45° to the $x, y$ axes ($E$ resonance eigenstate). For highlighting the integrated polymer, the raw SEM image is superimposed with the SEM image of the (same bare) nanocube taken before photopolymerization. Additional raw SEM images of hybrid nanocubes illustrating the good control of the fabrication procedure are provided as Supplementary Fig. 1. Figure 2b shows the corresponding calculated field modulus at a wavelength $\lambda = 780$ nm in a homogeneous refractive index $n = 1.50$ composed of both substrate and polymer formulation. This field is at the origin of the formation of the hybrid nanostructure. Figure 2c, d are, respectively, the SEM image of the resulting hybrid nanocube and corresponding calculated field modulus for an incident polarization parallel to the $x$ axis. In this case, plasmonic 2-photon polymerization turns out to yield a nanoscale polymer molding of the near-field of degenerated plasmonic dipolar modes, i.e., a linear combination of both in-plane resonance eigenstates of $E$ symmetry with relative weights determined by projections of the incident polarization on nanocube diagonals. This result constitutes an important new step forward compared to ref. [57] that deals with plasmonic 1-photon photopolymerization on a single nanocube and ref. [58] that demonstrated 2-photon polymerization of cationic system (SU8) in gap of pairs of gold nanocubes.

So, statistically, the polymer distribution follows the particle dipolar near-field distribution imposed and controlled by the incident polarization. The polymer distribution depends on the incident dose too (Fig. 2e; Supplementary Fig. 1). Figure 2a, c confirm that the presence of QDs inside the polymer does not

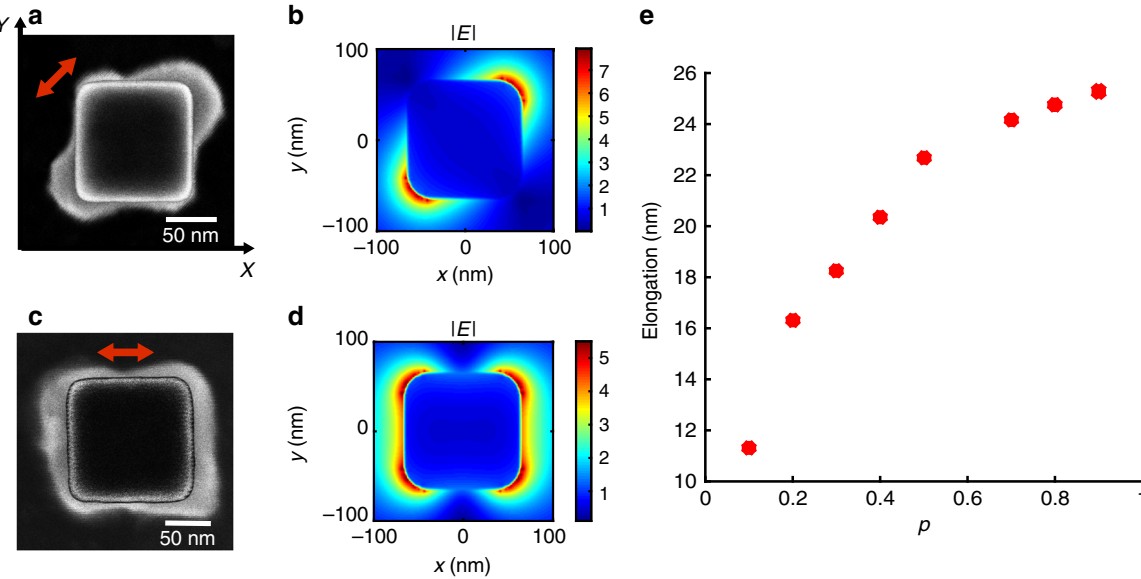

**Fig. 2 Au nanocube-based hybrid nanostructures made by plasmonic 2-photon polymerization. a** SEM image of the hybrid structure obtained with incident normal laser beam polarized along the cube diagonal (red arrow), $\lambda = 780$ nm. Image of the bare nanocube prior to photopolymerization is superimposed to highlight the polymer shell. **b** Finite difference time domain (FDTD) map of the field modulus |E| in the vicinity of the Au nanocube in polymer corresponding to case **a**, mid sectional horizontal plane, $\lambda = 780$ nm. **c** SEM image of the hybrid nanostructure obtained with incident polarization along the cube side edge (red arrow). Image of the bare nanocube prior to photopolymerization is superimposed to highlight the polymer shell. **d** FDTD map of the field modulus |E| in the vicinity of the Au nanocube in polymer corresponding to case **c**, mid sectional horizontal plane, $\lambda = 780$ nm. **e** Measured polymer elongation along the nanocube diagonal, case **a**, as a function of the normalized incident dose $p$ used for plasmonic 2-photon polymerization, as determined by SEM analysis.

prevent the 2-photon polymerization process[54]. The presence of QDs in the vicinity of a nanocube is difficult to detect by direct SEM imaging. This presence is ascertained by transmission electron microscopy (TEM) imaging (Supplementary Fig. 2c) and far-field observation of red PL from micropatterns fabricated by a laser writing technique based on 2-photon polymerization (Supplementary Fig. 2b).

In order to establish a quantitative link between the spatial in-plane extension of the polymerized lobes and the experienced local electromagnetic field, we carried out a parameter study in the case of Fig. 2a. Polymer elongation along the nanocube diagonal was measured for different values of $p$ ranging from 0.1 to 0.9. Figure 2e shows the result of the study. The polymer thickness increases as the dose increases in a nonlinear way. The apparent log-like function is the signature of the evanescent nature of the plasmonic field that triggered the polymerization process. This analysis, already reported in ref. [57] in the case of 1-photon plasmonic photopolymerization, is here extended to 2-photon polymerization as detailed in Supplementary Note 3. The analysis leads to the quantification of (i) the plasmon-induced intensity enhancement factor at the cube corner surface (=56, in close agreement to the value calculated by FDTD), (ii) the decay length (=7 nm, similar to that obtained from FDTD calculation) of the evanescent near-field. This shows that plasmon-induced 2-photon polymerization is an efficient method for quantitatively probing valuable plasmonic parameters that are difficult to get with other techniques like scanning near-field optical microscopy[59] and photoemission electron microscopy[60]. As far as the out-of-plane polymer extension is concerned, i.e., along the Z direction, differential AFM imaging (Supplementary note 4) showed that polymer gets integrated along the whole nanocube edge, resulting in a polymer nano volume slightly higher than the nanocube edge (about 135 nm high). As the polymer contains QDs (~a few tens of QDs for the biggest polymer lobes, see

Supplementary Note 2), the control of the polymer distribution shown in Fig. 2 offers a way to control $\rho(x, y, z)$, the spatial distribution probability defined in Eq. (1), making the hybrid system an anisotropic nano-emitter.

**Photoluminescence properties and overlap integral parametrization.** Figure 3 shows photoluminescence data ($\lambda_{\mathrm{em}} = 625$ nm) from single nanocube-based emitters ($\lambda_{\mathrm{exc}} = 405$ nm) fabricated with an incident polarization parallel to its diagonal (Fig. 2a case, cube E resonance eigenstate). It should be stressed that the 405-nm excitation wavelength has been chosen for efficient light absorption by the QDs. With regards to the plasmonic gold nanostructures this wavelength permits an off-resonant excitation, gold $5d$–$6sp$ interband transitions preclude any plasmon resonance for wavelengths below 520 nm. However, as it will be seen in Fig. 3, gold nanocube makes possible spatial confinement of the local field that excites the hybrid nanosource. Figure 3a shows the SEM image of the considered hybrid nano-object. Figure 3b, c show, respectively, the far-field PL image from a single hybrid nanosource and the corresponding PL spectrum. Under the same condition of excitation at 405 nm, no measurable emission at 625 nm was observed on bare nanocubes, bare polymer matrix or nanocubes exposed to a photosensitive formulation without any QDs. In addition, a similar PL spectrum was measured from a micronic pattern made of the same QD-containing polymer[54]. For comparison, hybrid polymer/gold nanocube structures without any QDs were produced and spectrally analyzed. No red PL emission was observed (see Supplementary Fig. 5b). These different observations demonstrate that the observed red PL comes from QDs trapped within the polymerized volume. Although one-day experiments did not reveal any significant reduction of PL intensity over time, a long-term study was carried out: the intensity of emission was regularly measured on four different hybrid nanosources for up to 25 days

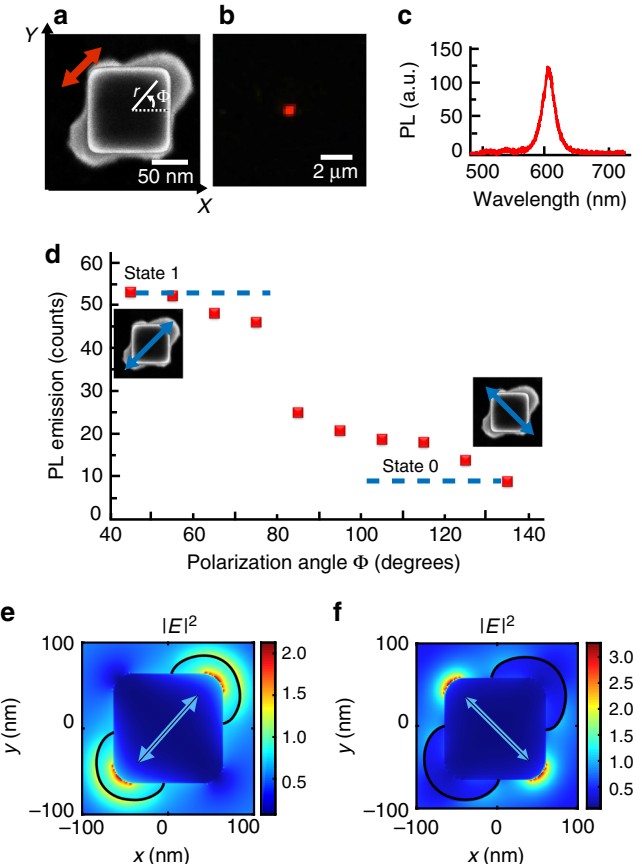

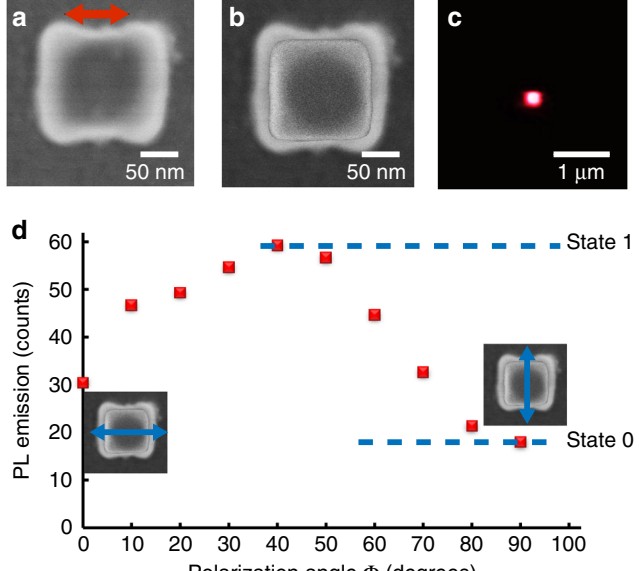

**Fig. 4 Photoluminescence measurements on single hybrid nano-emitters.**
Hybrid nanosource based on Au nanocube and CdSe/CdS/Zn quantum dots
embedded in polymerized volumes obtained under exciting polarization along
the cube edge side (red arrow). **a** Raw SEM image of the hybrid nanosystem.
**b** SEM raw image with superimposed initial bare nanocube. **c** Far-field PL
image at $\phi = 0°$ polarization angle, $\lambda = 405$ nm, linear polarization. **d** PL
intensity as a function of the angle of polarization $\phi$ of the blue exciting beam.
Blue arrows indicate two specific perpendicular polarizations.

**Fig. 3 Photoluminescence measurements on hybrid nano-emitters.** Single
hybrid nanosource based on a Au nanocube and CdSe/CdS/Zn quantum
dots embedded within nanometric polymer lobes obtained by plasmonic 2-
photon polymerization under $\phi = 45°$ linear excitation (red arrow). **a** SEM
image of the hybrid plasmonic nanosource with polar coordinates used for
discussion. The initial bare nanocube is superimposed to the raw SEM
image. **b** Far-field PL image at polarization angle $\phi = 45°$, $\lambda = 405$ nm,
linear polarization. **c** PL spectrum. **d** PL intensity as a function of the
angle of polarization of the exciting blue beam. Blue arrows indicate two
specific perpendicular polarizations **e, f** Calculated near-field intensity $|E|^2$,
$\lambda = 405$ nm, mid horizontal section planes, polarizations indicated by blue
arrows. Black lines represent the contours of the polymer lobes as deduced
from the SEM image (**a**).

(Supplementary Note 6). The signal stability observed over time
turns out to vary according to the incident power and the con-
sidered hybrid nanosources. Strongest nanosources presented
stable (~4% drop) PL for 5 days.

Figure 3a shows a clear anisotropy of the active medium
presenting a $C_{2v}$ in-plane symmetry with highly confined
distribution, suggesting a significant polarization sensitivity of
the emitter. We define $\rho(r, \phi)$, the probability of presence of the
nano-emitters as a function of polar coordinates $(r, \phi)$ represented
in Fig. 3a. From SEM images, $\rho$ is high for $(r \in [65–100$ nm$]) \cap$
$(\phi \in [20°–70°] \cup [200°–250°])$ and nil elsewhere (polymer layers
with thicknesses smaller than a QD diameter are unlikely to
contain QDs and are therefore neglected). As far as the azimuthal
angular distribution of nano-emitters is concerned, we define the
angular filling factor $\beta$, which quantifies the angular occupancy of
the active medium in the vicinity of the metal nanoparticle. In
Fig. 3a, the active medium occupies <30% ($\beta \sim 27\%$) of the space.
For comparison, the case of a spaser made of a spherical core-shell
plasmonic structure surrounded by a homogeneous layer of QDs[4]
exhibits an active medium in the region $(r \in [20–25$ nm$]) \cap (\phi \in$

$[0°–360°]$), i.e., an angular filling factor of $\beta = 100\%$. Figure 3d
shows the PL intensity of single hybrid nanosource as a function
of the incident polarization direction (excitation wavelength $\lambda =
405$ nm). The PL level varies quickly depending on the polariza-
tion direction. This effect results from the variation of the spatial
overlapping between the local near-field excitation and the
distribution of active medium. To illustrate this important point,
the near-field intensity at 405 nm was calculated by FDTD on a
realistic nanocube-based hybrid system presenting polymer lobes
at two cube corners. Figure 3e shows the exciting near-field for
incident polarization parallel to the polymer lobes. Although the
gold nanocube is non resonant at this wavelength, it acts as a
concentrator that confines light along the cube diagonal where the
QD presence probability $\rho(x, y, z)$ is high, resulting in a high level
of PL, named "state 1" in Fig. 3d. In Fig. 3e, apparent field
enhancement at the extremity of the polymer lobes is due to the
discontinuity of the field component perpendicular to the
polymer–air interface. In the case of an orthogonal polarization
(Fig. 3f), the near-field/active medium overlap is low, which
reduces the PL signal and corresponds to "state 0" in Fig. 3d. In
the latter case, the nanosource gets almost turned off, as shown in
Supplementary Fig. 7 that presents ten far-field PL images as a
function of incident polarization angles. It should be pointed out
that the quantum yield $Q$ is actually the effective quantum yield in
the presence of the metal nanoparticle. It is generally different
from the free space quantum yield and also depends on position
$(x, y, z)$[32]. In general, $Q$ decreases in the close vicinity (<10 nm) of
the particle due to non-radiative relaxation (quenching) and can
get strongly increased in the near-field of the plasmonic
nanoparticles[22]. In Fig. 3e, nano-emitters are expected to be
efficiently excited within the polymer (black contour) but
quenching can occur at the nanocube surface. Only QDs laying
on this surface (if there are any) are expected to be perturbed. In
Fig. 3f, the light is confined to the other two nanocube corners.
This configuration yields no emission since the probability of

presence of QDs at this specific location is nil (no polymer or negligible polymer thickness). So, the control of $\rho(r,\phi)$ constitutes a strong feature of this new type of hybrid nano-emitter and permits novel perspectives in hybrid nanoplasmonics.

**Polarization contrast.** PL polarization sensitivity can be discussed through a polarization contrast:

$$\delta_{\mathrm{PL}} = \frac{\mathrm{IPL}_{\mathrm{max}} - \mathrm{IPL}_{\mathrm{min}}}{\mathrm{IPL}_{\mathrm{max}} + \mathrm{IPL}_{\mathrm{min}}}, \qquad (3)$$

where $\mathrm{IPL}_{\mathrm{max}}$ and $\mathrm{IPL}_{\mathrm{min}}$ are, respectively, the maximum and minimum PL intensities measured after rotating the direction of the incident linear polarization. From Fig. 3d, we get $\delta_{\mathrm{PL}} \sim 0.7$. This high value is however limited by the PL background coming from the incident far-field whatever the polarization. In particular, in Fig. 3f, the calculated intensity of the exciting field is not nil within the black line contour that delimits the emitter-containing polymer, letting us expect a non-zero resulting PL. $\delta_{\mathrm{PL}}$ depends on the structure of the hybrid emitter that is controlled through proper choice of the metal nanoparticle geometry and the selection of plasmonic mode used for near-field polymerization. In order to illustrate this important possibility, different kinds of hybrid plasmonic nanosources were fabricated. Figure 4 shows the PL data from a single nanosource fabricated with an exciting field parallel to the cube edges. In this case, both diagonal plasmonic eigenmodes are symmetrically excited (off resonance excitation) and, from Fig. 2d, one expects all four corners to exhibit near-field enhancement. Figure 4a is the raw image of the hybrid nanosystem. Figure 4b is based on Fig. 4a and superimposes the initial cube to highlight the integrated polymer.

This nanosystem also presents a $C_{2v}$ in-plane symmetry. It is worth noting that this symmetry results from the $C_{4v}$ symmetry of the four hot spots shown in Fig. 2d. The weaker field excited at the cube sides along the $X$ axis (Fig. 2d) led to local polymerization at these sides (polymerization threshold was locally exceeded), resulting in a final $C_{2v}$ symmetry of the hybrid nano-object. This interesting example shows that nanoscale plasmonic photopolymerization can allow the control of the local degree of symmetry[55].

Compared to Fig. 3a, Fig. 4b presents a less confined polar distribution: $\rho$ is likely to be high for ($r \in [65-100\,\mathrm{nm}]$) ∩ ($\phi \in [295°-65°] \cup [115°-245°]$), corresponding to more than 70% of the angular space around the nanocube ($\beta \sim 72\%$). Figure 4c shows a typical PL far-field image of the hybrid nanosource. The PL signal (Fig. 4d) shows a weaker polarization dependence compared to the case in Fig. 3d. The highest PL level around 45° (225°) corresponds to thicker polymer volumes at the cube corners that results from electromagnetic singularities shown in Fig. 2d. PL contrast is measured to be $\delta_{\mathrm{PL}} \sim 0.3$.

**Other particle geometries.** The hybrid nanosystem presented in Figs. 2–4 were made from a cube with sharp corners and edges. For comparison, gold nanodisks[35] made by e-beam lithography were also considered for producing hybrid nanosources (Fig. 5). They have an initial $C_{\infty v}$ in-plane symmetry, a 90-nm diameter and are 50 nm thick. They present an in-plane dipolar plasmon resonance in air at 700 nm, permitting resonant plasmonic 2-photon polymerization under the same condition as those used for the nanocubes. In particular, incident linear 45° tilted polarization and dipolar near-field emission were used to get a 2-lobe hybrid nanosystem shown in Fig. 5a.

The fabricated structure presents two lobes along an axis that is 45° tilted relative to the $x$ axis, corresponding to the direction of polarization used during the plasmonic 2-photon-polymerization. The obtained active medium distribution is of $C_{2v}$ point group

symmetry and presents a pretty weak polar confinement: ($r \in [45-80\,\mathrm{nm}]$) ∩ ($\phi \in [5°-105°] \cup [185°-285°]$) corresponding to $\beta \sim 55\%$. Its PL polarization dependence (Fig. 5c) presents weak fluctuation with $\delta_{\mathrm{PL}} \sim 0.3$. Compared to the nanocube case, the dipolar near-field distribution leads to poorly confined photopolymerization. Here the $C_{\infty v}$ in-plane symmetry of the disk particle generates a dipolar near-field distribution with no sharp hot spots (no apexes)[35], yielding a low asymmetry of the final polymerized medium. It should also be pointed out that metal nanoparticles made by EBL can present, compared to chemically grown particles, local roughness and crystal defects that can result in a lower light confinement.

From the same gold nanodisk, an additional type of hybrid nanosource was made by using circular polarization at 780 nm for

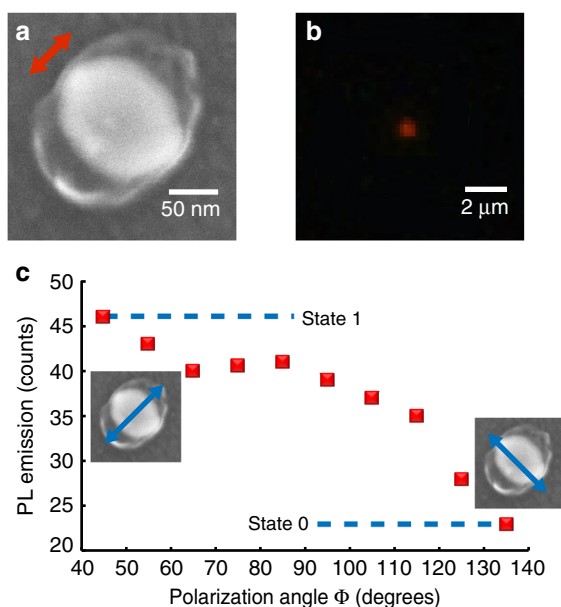

**Fig. 5 Photoluminescence measurements on single hybrid nano-emitters.** Hybrid nanosystem based on a Au nanodisk and CdSe/CdS/Zn quantum dots embedded within the polymer nanometric lobes. The hybrid nanosystem is obtained by plasmonic 2-photon polymerization using 45° tilted linear polarization (red arrow). **a** SEM image of the nanosystem (raw image). **b** Far-field PL image excitation at polarization angle $\phi = 45°$, $\lambda = 405\,\mathrm{nm}$, linear polarization. **c** PL intensity as function of the angle of polarization $\phi$, $\lambda = 405\,\mathrm{nm}$, linear polarization is represented as blue arrows.

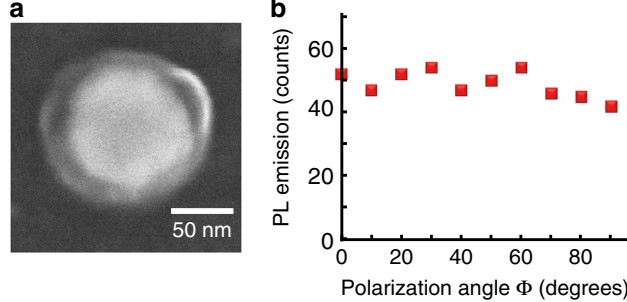

**Fig. 6 Photoluminescence measurements on single hybrid nano-emitters.** Hybrid nanosource based on Au nanodisk and CdSe/CdS/Zn quantum dots embedded in a nanometric polymer shell obtained by plasmonic 2-photon polymerization using circular polarization. **a** SEM image of the hybrid nanosource. **b** PL intensity as a function of the polarization angle $\phi$ of the blue excitation beam. $\lambda = 405\,\mathrm{nm}$, linear polarization.

plasmon-induced 2-photon polymerization (Fig. 6). The resulting nanosystem, shown in Fig. 6a, is characterized by an active medium exhibiting a ring like distribution of $C_{\infty v}$ symmetry, i.e., occupying $\beta = 100\%$ of the angular space. Accordingly, the PL varies poorly with the polarization angle of the exciting field (Fig. 6b). A low PL contrast $\delta_{PL} \sim 0.1$ is obtained from the experimental data. This slightly positive value reflects the imperfection of the circular pattern and the inhomogeneous distribution of QDs within the polymer volume.

**Overlap integral ratio.** Experimental data obtained on the previous four kinds of plasmonic hybrid nanosources can be discussed in terms of nanoscale spatial overlap integral between nano-emitters spatial distribution and local near-field configuration. Inspired by Eq. (2), we define a normalized spatial overlapping ratio $\eta_{nf/em}$ between the off-resonant exciting plasmonic near-field intensity and the distribution of QD emitters:

$$\eta_{nf/em} = \frac{V_0 \iiint E_{exc}^2 \times \rho \, dV}{\iiint E_{exc}^2 \, dV \times \iiint \rho \, dV}, \quad (4)$$

where $E_{exc}(x, y, z)$ is the modulus of the local plasmonic field that excites the QDs at 405 nm wavelength. It is calculated by FDTD. $\rho(x, y, z)$ is the volume density of probability of presence of the nano-emitters, as defined in Eq. (1). It can be assessed experimentally from SEM and AFM images. $V_0$ is a constant that is homogeneous to a volume. It can be considered as the total volume of integration.

$\eta_{nf/em}$ quantifies the way the local exciting field intensity and the nano-emitters distribution spatially overlap with each other for a given situation, with a given excitation polarization direction. For example, $\eta_{nf/em} = 0$ means that overlap is nil: QDs do not get excited at all and resulting PL is expected to be negligible. On the other hand, $\eta_{nf/em} = 1$ leads to the highest possible PL. The most important part of Eq. (4), in terms of physical meaning for describing the overlap, is the numerator. The denominator is only used for normalization. For any given hybrid nanostructure excited with a given polarization direction this denominator has a constant positive value and never goes to zero (the integral $\iiint \rho \, dV$ is always strictly positive, although some elements $\rho \, dV$ within the integral corresponding to an absence of polymer can be locally nil).

Figure 7 shows the calculated $\eta_{nf/em}$ ratio as a function of the polarization direction of the incident field at 405 nm, for three different types of hybrid nanosources. It should be reminded that a given polarization direction corresponds to a specific $E_{exc}(x, y, z)$ spatial distribution while $\rho(x, y, z)$ is fixed for a given hybrid nanostructure. For this calculation, we considered that both orientation and spatial distribution of QDs within the polymer matrix are random and do not change during excitation. For the sake of simplicity, in the presence of polymer $\rho = 1$, in the absence of polymer $\rho = 0$. In other words, the $\rho(x, y, z)$ map is a homogeneous reproduction of the polymer distribution in the vicinity of the metal nano-object and the QD distribution within the polymer is assumed homogeneous.

From Fig. 7, it turns out that overlap integral $\eta_{nf/em}$ varies like the PL intensity (Figs. 3–5), showing that the PL level directly depends on $\eta_{nf/em}$. For instance, the maximum at 45° in Fig. 7b corresponds to the high polymer thickness along the cube diagonal, as observed experimentally in Fig. 4d. Each value of $\eta_{nf/em}$ is associated to a PL intensity value, so the overlap integral $\eta_{nf/em}$ is an important control parameter. The link between $\eta_{nf/em}$ and IPL (the PL intensity) can be formally established. We consider that IPL results from the PL issued from an ensemble of volume elements $dV$ in the vicinity of the metal nanostructure.

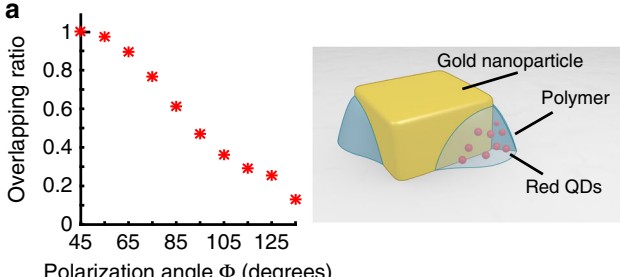

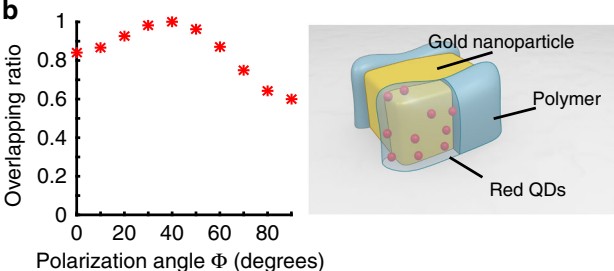

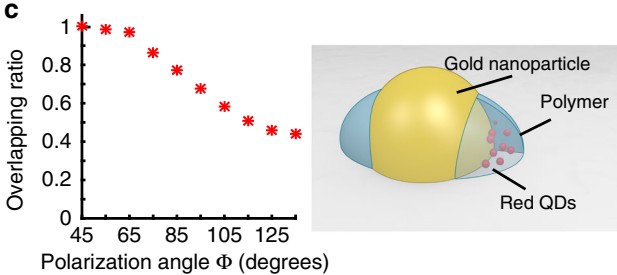

**Fig. 7 Overlap integral ratio.** Computed spatial overlap integral $\eta_{nf/em}$, defined in Eq. (4), between the local excitation field and the active medium as a function of the incident polarization angle $\phi$ for three different hybrid nanosources. The incident field is linearly polarized, $\lambda = 405$ nm. **a** Au nanocube with polymer lobes along the diagonal direction, $\phi = 45°$. **b** Au nanocube with polymer deposits at cube faces, $\phi = 0°$. **c** Hybrid Au nanodisk with polymer lobes along $\phi = 45°$.

Each volume element, positioned at $(x, y, z)$, emits a PL intensity $d_{IPL}$ that is defined as:

$$d_{IPL} = \alpha \times \gamma_{exc}(x, y, z, \nu_{exc}) \times Q(\nu_{em}) \times \rho(x, y, z) dV, \quad (5)$$

where $\alpha$ is a constant including the incident intensity and the efficiency of light collection (setup geometry, numerical apertures of lenses, see the "Methods" section). The other parameters of Eq. (5) have already been defined in Eq. (1). As a reminder, $\rho(x, y, z)dV$ is the probability of presence of emitters inside $dV$. For a given hybrid metal particle, we assume that $\alpha$ and $Q(\nu_{em})$ are constant in Eq. (5). In particular, it is supposed that the PL results from an average constant effective quantum yield, averaging out quenched nano-emitters touching the metal surface. $\gamma_{exc}$ is proportional to $|\mathbf{E}_{exc} \cdot \mathbf{\mu}|^2$, the square of the scalar product between the local exciting field vector and the QD transition dipole moment $\mathbf{\mu}$[61]. By considering a statically constant dipole moment, $\gamma_{exc}$ becomes proportional to $E_{exc}^2$ and $d_{IPL} \propto E_{exc}^2 \times \rho \, dV$. The resulting IPL signal can thus be expressed as:

$$IPL = \iiint d_{IPL} \propto \iiint E_{exc}^2 \times \rho \, dv. \quad (6)$$

From Eqs. (4) and (6), it turns out that IPL is proportional to $\eta_{nf/em}$. The two quantities are thus clearly connected to each

other and Eq. (3) can be rewritten as

$$\delta_{PL} = \frac{IPL_{max} - IPL_{min}}{IPL_{max} + IPL_{min}} = \frac{\eta_{nf/em}^{max} - \eta_{nf/em}^{min}}{\eta_{nf/em}^{max} + \eta_{nf/em}^{min}} = \delta_{nf/em}, \quad (7)$$

where $\eta_{nf/em}^{max}$ and $\eta_{nf/em}^{min}$ are, respectively, the maximum and minimum value of $\eta_{nf/em}$. The third term of Eq. (7) $\delta_{nf/em}$ calculated from Fig. 7 data gives 0.74 (Fig. 7a), 0.25 (Fig. 7b), and 0.35 (Fig. 7c). These values can be compared with those of $\delta_{PL}$ that were experimentally determined: 0.7, 0.3, and 0.3, respectively. As predicted by Eq. (7), it turns out that $\delta_{nf/em}$ and $\delta_{PL}$ are equal. This important result validates the proportionality link between $\eta_{nf/em}$ and IPL, although local QD inhomogeneity could explain some unexpected fluctuation in the PL plot (e.g., Figs. 5c and 6b).

**Toward a single-photon switchable hybrid nano-emitter.** The possibility of controlling the nanoscale spatial overlap between an exciting field and the active medium was extended to the single photon regime. The concentration of QDs within the photo-polymerizable formulation was decreased using the method described in the Methods section, allowing us to trap a small number of QDs (a single QD or a few ones) inside the polymer lobes of a nanocube-based hybrid emitter. Figure 8a shows an AFM image of such a hybrid emitter that is similar to that presented in Fig. 2a. The corresponding PL spectrum was obtained using a home-build micro-PL setup sensitive to single quantum emitter emission (see "Methods" section and Supplementary Fig. 8). We can see a clear blinking from Fig. 8b, c which is the signature of single or few QDs emission (time trace 50 s, excitation laser wavelength 405 nm, incident polarization parallel to the polymer lobe along the $x$ axis). More interestingly, this emission gets switched off when the incident polarization is rotated to 90° (Fig. 8b) due to the sudden lack of overlap between the exciting near-field and the single QD. This constitutes a first demonstration of a polarization-driven switchable single photon emission. Auto-correlation function $g^{(2)}$ measurement was carried out to confirm and determine the photon emission regime. Principle of the $g^{(2)}$ measure is reminded in the Methods section. We found that at a zero delay, $g^{(2)}(0) \sim 0.35$ (Fig. 8d), i.e., below 0.5, which is the signature of a single photon emission[62]. This result was also obtained on single QDs in polymer without gold nanocubes (Supplementary Fig. 9b). We determined the Purcell effect due to the weak coupling between the trapped single QD and the gold nanocube. The instrument response function (IRF) of the system is measured every time before the lifetime measurement of the hybrid nano-emitter and amounts to 0.63 ns (see "Methods" section and Supplementary Note 8). We measured a lifetime of 0.725 ns (Fig. 8e) on a typical hybrid nano-emitter, whereas the lifetime of single QDs in polymer are measured to be around 17.5 ns (Fig. 8f, red curve, and Supplementary Fig. 9b). These measures correspond to an averaged Purcell factor of (17.5/0.725) = 24. Statistically, several hybrid nanosources revealed smaller lifetimes, close or probably smaller than the IRF (see for example yellow curve in Fig. 8f), suggesting higher Purcell factors (see Supplementary Note 9), larger than 28 (=17.5/0.63). This variation of lifetime is believed to be related to the random position of the QD within the polymer lobes in the vicinity of the gold nanocube: farthest QDs present a lifetime of ~0.8 ns, whereas closest ones have a lifetime close and even below the resolution of our system.

In conclusion, different hybrid plasmonic nano-emitters have been fabricated by plasmon-based 2-photon nanoscale polymerization. The hybrid emitters present anisotropic spatial distribution of the active medium with different controllable degrees of symmetry. The resulting polarization dependence of the photo-luminescence has been analyzed and quantified on the basis of

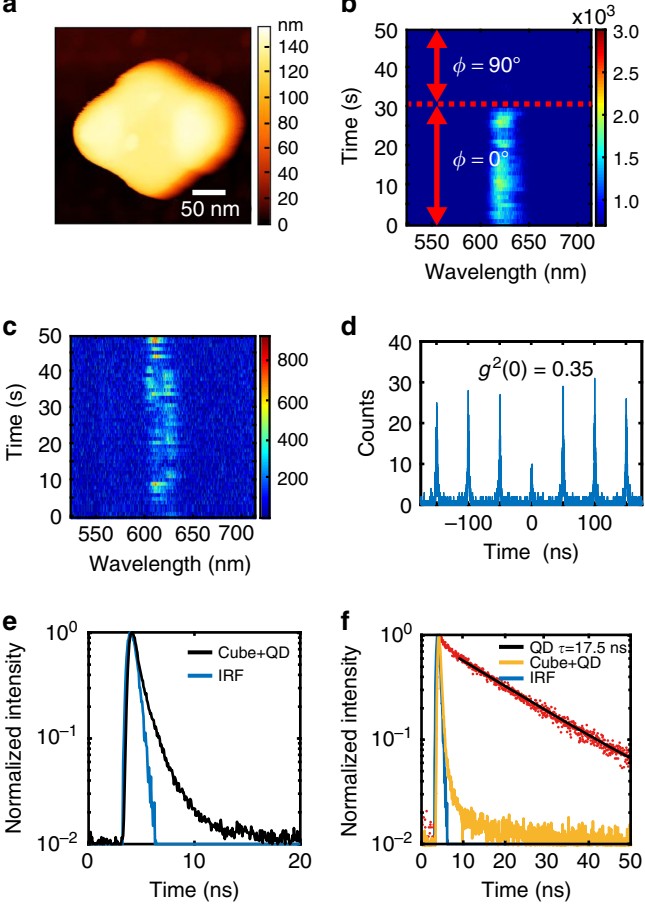

**Fig. 8 Hybrid nano-emitters in the single photon regime. a** AFM image of a nanocube-based hybrid nano-emitter. The polymer lobes contain a single or a few QDs. **b**, **c** PL spectrum time trace of $t = 50$ s excited by linearly polarized laser along $\phi = 0°$ at 405 nm. In **b**, at time $t = 32$ s, the polarization direction is rotated to $\phi = 90°$. **d** $g^{(2)}$ measurement showing $g^{(2)}(0) = 0.35$. **e** Typical lifetime measurement on a single-QD hybrid nano-emitter, lifetime ~0.725 ns. The blue curve represents the instrumentation response function of 0.63 ns. **f** Lifetime measurements. Comparison between single QD in polymer without gold nanocubes (measure: red curve, fitting: black curve 17.5 ns) and single QD in the vicinity of a gold nanocube (hybrid nano-emitter: yellow curve).

new specific parameters (i) spatial distribution of nano-emitters including angular filling factor of the active medium, (ii) nanoscale spatial overlap integral between active medium and exciting near-field, and (iii) associated photoluminescence polarization contrast. In summary, we presented an approach that ensures a good overlap between the location of the emitters and the maxima of the electric field. It is unlikely that other methods[41–47,49,50] make possible the positioning of quantum emitters with so diversified degrees of symmetry. This is due to the fact that our approach has a major asset: it uses the intrinsic plasmonic field to position the quantum emitters through local plasmonic photopolymerization. These hybrid systems can emit in a single-photon regime. A preliminary result of a polarization-driven single photon switch is reported.

This new class of anisotropic plasmonic nano-emitters opens up the avenue for polarization-driven tunable nano-emitters including nanolasers and single-photon emitters. In particular, regarding nanolasers, it is reasonable to assume that the effective pumping intensity is directly dependent on the overlap integral $\eta_{nf/em}$. Polarization could thus command both the pumping

process and the associated color in the case of anisotropic nanoscale distribution of differently colored nano-emitters[52].

## Methods

**Synthesis of nanocubes**. 127 nm Au-cubes were synthesized in aqueous solution in presence of the cetyltrimethylammonium bromide surfactant, following the seed mediated growth protocol described in detail in ref. [53]. Small nanocubes of 40 ± 2 nm were synthesized first, and then used as "seed" in a second step of growth. The growth solution (5 ml) was prepared by successively mixing 3.6 ml of a 22 mM CTAB solution with 100 μl of $HAuCl_4$ (0.01 M) and 1.3 ml of ascorbic acid (0.01 M). Adding the seeds (100 μl) initiates the growth of the cubes. Sharp edges and corners were obtained by adjusting the ratio of ascorbic acid to $HAuCl_4$ to a value of 13. After one night at 30 °C, 127 nm nanocubes roughly purified (~60%) were obtained, through selective sedimentation at the bottom of the reaction vessel owing to a depletion-induced interaction. The solution was then centrifuged twice and redispersed in water to remove excess CTAB down to ~1 mM. The chemicals used were cetyl-trimethylammonium bromide (CTAB ≧ 98%), chloroauric acid ($HAuCl_4 \cdot 3H_2O$), sodium borohydride ($NaBH_4$, 99%), and ascorbic acid (99%). They were purchased from Sigma and used as received. Deionized water is used for all experiments.

**Sample preparation**. The substrates are glass coverslips, 150-μm thick, covered with a conducting 80-nm thick indium-tin-oxide (ITO) layer (SOLEMS, Palaiseau, France). Before deposition, ITO substrates were exposed during 20 min to an UV ozone lamp to make the surface hydrophilic. Right after, 2 μl of a water-diluted solution of nanocubes (~3.10$^{-13}$ M of cubes) was deposited and left to dry. After drying, the sample was copiously washed with ethanol, dried and then cleaned by UV/Ozone to remove any organic residues. This method leads to an averaged surface density of 1 nanocube/10 μm$^2$.

**Characterization of the gold nanocubes**. Figure 1a shows a SEM image of a deposited single gold nanocube. Figure 1b shows the corresponding AFM image. The colloidal particle is of well-defined shape and no signature of residual CTAB surfactant is visible. By SEM, we studied a hundred identified/labeled cubes, their size distribution is reported in Fig. 1c. As the statistical chart shows, the cube sizes are distributed according to a Gaussian-like distribution with a main average edge size of about 127 ± 2 nm. Good knowledge of this value and related tiny Gaussian distribution make possible parameter analysis and make us confident in the parameters used for electromagnetic calculation.

Figure 1d is a scattering dark-field image (condenser NA = 0.9, objective NA = 0.6) showing that it is possible to address specific single gold nanocube (typical density is 0.1 cube μm$^{-2}$). The spectrometer is coupled to an inverted optical microscope (Olympus X71) on which an AFM head (Veeco Bioscope 2) is installed, allowing us to easily correlate morphology and optical spectrum of identified nanocubes and discriminate between pairs (or aggregates) and single particles. Figure 1e is the experimental scattering spectrum of a single specific nanocube. It presents a main peak at 680 nm that is attributed to in-plane dipolar mode, whereas the shoulder around 525 nm is attributed to a higher order mode. These peaks where retrieved by FDTD calculation with gold nanocubes on ITO-coated glass substrate in air (blue curve in Fig. 1f).

**Formulation preparation and threshold determination**. The red QDs were synthesized by a solution process. (i) a three-neck flask loaded cadmium oxide (1 mmol), zinc acetate (4 mmol), oleic acid (5 ml), and 1-octodecene (20 ml) and filled with $N_2$ was heated at 150 °C for 1 h followed by 300 °C for 30 min to form a clear solution; (ii) a mixture of trioctylphosphine (TOP 0.25 ml) and selenium (0.25 mmol) was quickly injected into the flask and temperature was maintained at 300 °C for 1.5 min to form the core of CdSe; (iii) dodecanethiol (0.2 ml) was added slowly to the reactor and the reaction was kept at 300 °C for 30 min to form the inner shell of CdS; (iv) sulfur (4 mmol) dissolved in TOP (2 ml) was added quickly to the solution at 300 °C for 10 min to form the outer shell of ZnS. Then the synthesized QDs were purified by ethanol followed by centrifugation. The process was repeated 5 times and finally pure QDs were obtained. Then the synthesized QDs (10 mg) were dissolved in hexane (10 ml). A multifunctional monomer mixture based on PETA (2.5 ml) was then added to the solution followed by a vigorous shaking for 15 min for completing the ligand exchange process. The phase separation solution was placed in a vacuum oven overnight to remove the hexane completely. Finally, the photopolymerizable formulation was finished after adding 1%wt Irgacure 819 into the QDs-grafted PETA solution and mixing for 30 min. The average diameter of CdSe/CdS/Zn QD particles was 6.6 nm ± 0.7 nm[63].

The threshold $D_{th}$ of incident energy for 2-photon polymerization is identified by the exposure time and the laser intensity per unit area. In our case, polymerization was achieved on the Olympus IX71 sample plate. The femtosecond laser at $\lambda$ = 780 nm was focused by a lens with NA of 0.6 on a clean ITO-coated glass substrate without any particles. During all of the exposure period, the light path was kept unchanged. Although the optical shutter was kept at 125 ms, the laser intensity was decreased step by step, meanwhile, the size of the polymer dot got smaller and smaller. The smallest intensity before polymer dot disappeared was

chosen as the threshold intensity. In order to obtain single (or few) QDs hybrid nano-emitters, QD concentration within the photosensitive formulation was decreased. 4 mg IRG819 was added into 396 mg PETA to prepare a photosensitive diluent of 400 mg, which was mixed by the magnetic mixer for 30 min with a temperature of 30 °C. 100 mg of the original formulation (PETA + QDs + 1% IRG819) and the diluent were mixed together and stirred until homogenization. In that way, we got a low concentration (20% of the original) formulation. The original formulation contains 1 mg QDs in 1 g polymer, whereas the low concentration formulation contains 0.2 mg QDs in 1 g polymer.

**Procedure for fabricating the hybrid nano-emitter**. In order to avoid any changes in light path and polarization, after threshold energy was found, we placed the cube deposited ITO-coated substrate in the focus plane directly. Each isolated single cube was then moved in the laser focus area by a stepper motor stage and exposed at an intensity below the threshold. A drop of photosensitive formulation was deposited on a pre-identified gold nanocube. A Ti:Sa pulsed femtosecond laser ($\lambda$ = 780 nm) was used as the excitation light, which fits IRG819 2-photon absorption (IRG819 presents strong one-photon absorption in the 350–450 nm range) and lies inside the nanocube in-plane dipolar plasmon resonance in the presence of the polymerizable solution (see calculated plasmon spectrum Fig. 1f, orange curve). Incident light was focused (1.6 μm spot, NA objective lens = 0.6) from under the substrate onto the substrate top surface, allowing one to address single pre-identified gold nanocubes. The incident field was in-plane and linearly polarized. It should be pointed out that the relatively low numerical aperture of the objective lens used for light focusing (0.6) keeps the linear polarization parallel to the sample plane. Exposure time was 125 ms and average incident power at the sample plane was in the [40-400] μW range, resp. [0.5–5.0 kW cm$^{-2}$], permitting to adjust the incident dose $D_{in} = p \cdot D_{th}$, where $p < 1$.

After exposure, the sample was rinsed with acetone, hydrochloric acid solution, and isopropanol, each process lasting 10 min. Excess of QDs and liquid formulation was efficiently removed from the substrate by this process.

**PL measurement on hybrid nano-emitters**. The signal was analyzed by a spectrometer coupled to an inverted optical microscope (Olympus IX71). We used a 405 nm continuous laser to excite fluorescence. The signal was then collected by a ×50 objective lens (NA = 0.8) used also for focusing the 405-nm excitation light onto single hybrid nano-emitters, and separated from the laser excitation using a 514 nm long pass filter. For single nano-emitter measurements, image mode is used with the slit fully open and the laser spot was firstly modified to the center and was marked on the CCD image. After aligning the targeted single nanoparticle to the mark, spatial filtering is made through adjustment of the slit size to detect an area less than 1 μm × 1 μm, ensuring single object signal measurements. The polarization direction was changed by a half-wave plate and checked by a linear polarizer. The excitation power was kept at 2 μW μm$^{-2}$ (50 s exposure time) to obtain a single fluorescence spectrum. For fluorescence images acquisition, 60 s exposure time and 20 μW μm$^{-2}$ incident power were used.

**Analysis of single-QD hybrid plasmonic nanosources**. A dedicated micro-PL setup described in Supplementary Note 8 was used. The emitted light from single hybrid nano-emitters was redirected toward a single photon detector (Si avalanche photodiode, APD, from EG&G) for lifetime measurement. A ×100/0.95 objective lens was used for efficient single photon collection. We used a pulsed 405-nm laser diode with repetition frequencies of 20 and 5 MHz for $g^{(2)}$ and lifetime measurement, respectively. The IRF was measured on a pure clean glass substrate as a reference sample. We obtained an IRF of about 0.63 ns. For lifetime measurements on the hybrid plasmonic nanostructure, we consider that, at the time being, our time resolution is 0.63 ns but further studies will be using the fact that, in principle, lifetimes down to 1/10 of the IRF width (FWHM) can still be recovered via iterative reconvolution (Supplementary Note 9). For $g^{(2)}$ measurement, within the same part of the setup, collected light from the sample was split into two beams that were coupled into optical fibers. These two fibers were connected to two APD detectors in order to measure the photon antibunching behavior of the $g^{(2)}$ autocorrelation function. This so-called Hanbury Brown and Twiss configuration allowed us to measure single photon emitters with great accuracy (see Supplementary Fig. 9b for $g^{(2)}$ measurements on single QDs in polymer).

**FDTD calculations**. The FDTD calculations were carried out with a commercial software: *FDTD Solutions* from Lumerical (cf. https://www.lumerical.com/products/fdtd/). Yee cell size was $1 \times 1 \times 1$ nm$^3$.

## Data availability

The authors declare that all data supporting the findings of this study are available within the paper [and its supplementary information files].

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

## Acknowledgements

Experiments were carried out within the Nanomat platform (www.nanomat.eu) supported by the Ministère de l'Enseignement Supérieur et de la Recherche, the Région Grand Est, the Conseil Général de l'Aube, and FEDER funds from the European Community. This project is financially supported by the ANR (French Research Agency) and the NRF (National

Research Foundation, Singapore) through the International ACTIVE-NANOPHOT (alias "MULHYN") funded project (ANR-15-CE24-0036-01 and NRF2015-NRF-ANR000-MULHYN). FEDER (Fonds européens de développement régional) and Graduate school EIPHI-BFC (ANR-17-EURE-0002) are acknowledged for their funding support of A. Issa's postdoctoral fellowship (Nano-integration project). R. Bachelot thanks Loic Le Cunff for providing illustration for Fig. 7 (right-hand column) and Farid Kameche for his participation in the first tests of plasmonic photopolymerization on TEM grids. D. Ge thanks the CSC for funding support. M. Nahra and C. Couteau are thankful to the European Union's Horizon 2020 Research and Innovation Program under Marie Sklodowska-Curie Grant Agreement No. 765075 (LIMQUET). T. Xu acknowledges the support of the National Natural Science Foundation of China (Grant No. 61775130). This work has been made within the frame of the Graduate School (Ecole Universitaire de Recherche) "NANO-PHOT", contract ANR-18-EURE-0013.

## Author contributions

D.G. carried out the experiments, performed the data acquisition, performed the FDTD calculation, and analyzed the data. S.M. produced the gold nanocubes by chemical synthesis and took part in the design of the studies, data interpretation, and analysis. J.B. produced the gold nanodisks by e-beam lithography. R.D. and M.N. provided help in the development and use of instrumentation for the $g^{(2)}$ and lifetime measurements at the single nano-emitter level. A.I., S.J., T.H.N., X.Q.D., and X.Y. developed QD-containing photopolymers. H.C. took part in the measurements. S.B., C.C., J.P. C.F., L.D., O.S, X.W.S., C.D., T.X., B.W., and R.B. took part in the design of the studies, data interpretation, and analysis. C.C. led the implementation at the single photon regime. R.B. is the project PI. He conducted the research project and wrote the manuscript.

## Competing interests

The authors declare no competing interests.
