## [Peer Review File · Nature Communications]

Reviewers' comments:

Reviewer #1 (Remarks to the Author):

The manuscript reports the fabrication of hybrid nanoemitters made of quantum dots distributed over metallic nanoparticles. The originality of the paper is to manage to control the position of the emitters in certain areas around the metallic nanoparticles. This is achieved by embedding the emitters in a polymer which is then illuminated by the local field around the particle. After development, the QDs remains only where the local field was larger than a polymerization threshold. As a result, the hybrid nanoparticle has non-isotropic absorption and emission properties.

This is an interesting paper that I recommend for publication after revision. I have a number of remarks and questions.

1. The main concern is to improve readability of the manuscript.

a. It contains at least 15 acronyms (HPN, MNP, NE, LDOS, PL, GNC, ITO, CTAB, SEM, AFM, FDTD, PETA, 2PP, QD, TEM). I acknowledge that some of them are well known and that molecules need to be named. Nonetheless, 2PP, HPN, MNP, NE, GNC are not very helpful. It makes the reading difficult so that the number should be reduced.

The manuscript reports the fabrication of hybrid nanoemitters made of quantum dots distributed over metallic nanoparticles. The originality of the paper is to manage to control the position of the emitters in certain areas around the metallic nanoparticles. This is achieved by embedding the emitters in a polymer which is then illuminated by the local field around the particle. After development, the QDs remains only where the local field was larger than a polymerization threshold. As a result, the hybrid nanoparticle has non-isotropic absorption and emission properties.

This is an interesting paper that I recommend for publication after revision. I have a number of remarks and questions.

1. The main concern is to improve readability of the manuscript.

a. It contains at least 15 acronyms (HPN, MNP, NE, LDOS, PL, GNC, ITO, CTAB, SEM, AFM, FDTD, PETA, 2PP, QD, TEM). I acknowledge that some of them are well known and that molecules need to be named. Nonetheless, 2PP, HPN, MNP, NE, GNC makes the reading difficult so that the number should be reduced.

b. I recommend to shorten the paper. Many details should be moved to methods or supplementary.

c. There are sentences that should be corrected:

The resulting polarization dependence of the photoluminescence has been analyzed and quantified on the bases new specific parameters whose use and definition

Compared to the nanocube case, the dipolar near field distribution yields of 2PP process with reduced control over.

2. The discussion of the parameter η (eq. (4)) is not convincing and should be better explained and demonstrated or removed. On one hand, Fig. (7) does not compare on the same graph this quantity with the data which are in Fig 3-4. There is only a trend agreement. On the other hand, the introduction of this quantity is not motivated. If it is expected that it varies like the signal, it should be derived starting from Eq. (1).

The same remark applies to δ' . It should be specified that max and min are defined when η varies as a function of angle (presumably). It is unclear if the agreement between δ and δ' is accidental or whether it is expected on the basis of some physical arguments. Without a physical argument predicting this behaviour, in my opinion, there are not enough evidences to draw any useful conclusion at this stage.

Reviewer #2 (Remarks to the Author):

The manuscript presents the preparation and characterization of hybrid light emitters based on plasmonic particles and quantum dots. The fabrication method is based on plasmon-enhanced two-photon polymerization and enables the authors to position the quantum dots at different positions around the metal particles. For optical characterization, two quantities are defined: the spatial overlap integral between emitters and the plasmonic field, and an excitation polarization contrast.

Positioning emitters near plasmonic particles is not novel, as the authors also acknowledge in the introduction. Several methods were developed over the years: chemical linking, AFM-based positioning, lithography methods, or DNA assembly (not cited by the authors). Part of the authors developed over the last decade methods based on photopolymerization. Compared to previous demonstrations based on photopolymerization, I am afraid that the work could be considered incremental. The possible novelty then focuses mostly on analyzing the spatial overlap integral for single nanoparticles, but still it does not provide new knowledge gain. There are no significant technical issues, but I believe that the article will not be influential enough in the (sub-)field to warrant publication in Nature Communications.

The manuscript might benefit from attention to the following points:

- Equation 1 seems to neglect Stokes shift (wavelength dependence of excitation and emission).
- Please substantiate the claim that "the HPNs are stable over time. In particular, they do not suffer from photobleaching" with data. How long were the HPNs exposed to light to reach this conclusion? How? Please quantify it.
- Can the authors at least estimate the number of quantum dots attached to each nanocube?
- Figure 4b: please describe better the meaning of this image composition, with more accuracy, as it is open to interpretation. Is it precisely the same GNC that is covered with polymer in Figure 4a? It is strange to merge two SEM images instead of highlighting the profile of the GNC on the final nanostructure.

Ps: There is an incomplete sentence: "Compared to the nanocube case, the dipolar near field distribution yields of 2PP process with reduced control over".

Renaud Bachelot
Laboratory Light, nanomaterials, nanotechnology (L2n)
CNRS, ERL 7004
phone : +33 3 25 71 56 65
fax : +33 3 25 71 84 56
E-mail: renaud.bachelot@utt.fr

REVISED MANUSCRIPT NCOMMS 19-33496

Response to referees

We have considered with great care the recommendations of both referees that we thank for having taken time to consider our article.

We were glad to see that the Reviewer 1 considered our manuscript as interesting and recommended its publication in *Nature Communications*.

In contrast, Reviewer 2 thinks that the paper is not original enough, as compared to previous publications on plasmonic photo polymerization.

We believe this new version provide an answer to all of the reviewer's criticisms. We also believe that, contrary to the reviewer 2's opinion, our results are original and provide significant new data and information on hybrid plasmonic nano-emitters, including the related physics behind. It is actually not easy, as authors, to provide an answer to such a clear-cut personal point of view. In order to address the reviewer 2' concern, we have i) highlighted the novelties throughout the manuscript, ii) completed the analysis of the overlap integral that is now analytically linked to the photoluminescence intensity (as suggested by reviewer 1), iii) addressed all the points raised by reviewer 2 (with new data), iv) decided to include new recent results on switchable single-photon emission from hybrid nano-emitters based on gold nanocubes. Initially, we planned to prepare a new dedicated article to present this important achievement but partial integration of these new results into our *Nature Communications* paper will certainly make it stronger in terms of novelties and impact.

Reviewer #1 (Remarks to the Author):

The manuscript reports the fabrication of hybrid nanoemitters made of quantum dots distributed over metallic nanoparticles. The originality of the paper is to manage to control the position of the emitters in certain areas around the metallic nanoparticles. This is achieved by embedding the emitters in a polymer which is then illuminated by the local field around the particle. After development, the QDs remains only where the local field was larger than a polymerization threshold. As a result, the hybrid nanoparticle has non-isotropic absorption and emission properties.

This is an interesting paper that I recommend for publication after revision. I have a number of remarks and questions.

We thank reviewer 1 for this positive opinion and for the very useful following remarks and questions

1. The main concern is to improve readability of the manuscript.
a. It contains at least 15 acronyms (HPN, MNP, NE, LDOS, PL, GNC, ITO, CTAB, SEM, AFM, FDTD, PETA, 2PP, QD, TEM). I acknowledge that some of them are well known and that molecules need to be named. Nonetheless, 2PP, HPN, MNP, NE, GNC are not very helpful. It makes the reading difficult so that the number should be reduced.

→ as suggested, we kept well known acronyms and removed the following acronyms: 2PP, HPN, MNP, NE, GNC, GND (see **yellow parts** in the revised manuscript)

b. I recommend to shorten the paper. Many details should be moved to methods or supplementary.

Sections concerning the nanocube characterization and process of plasmon induced 2-photon polymerization have been transferred into the “method” section

→ see transferred **yellow part** in the “method” section

c. There are sentences that should be corrected:
The resulting polarization dependence of the photoluminescence has been analyzed and quantified on the bases new specific parameters whose use and definition

Compared to the nanocube case, the dipolar near field distribution yields of 2PP process with reduced control over.

The two sentences have been corrected. In general, the article has been deeply reviewed in order to remove all the residual typos

→ see **yellow parts** in the revised manuscript

2. The discussion of the parameter eta (eq. (4)) is not convincing and should be better explained and demonstrated or removed.

Eta (η) is actually an important new parameter that is introduced here for the first time within the context of hybrid nanoplasmonics. Its name has been modified ($\eta \rightarrow \eta_{nf/em}$) to avoid any confusion with Eq. 2 that inspired the definition of $\eta_{nf/em}$. $\eta_{nf/em}$ actually allows for the quantification of the spatial overlap between the exciting optical local near-field and the active medium which contains nano-emitters. So far, this parameter could not be defined because the spatial distribution was not really controlled at the nanoscale. In the case of our study, controlling this spatial distribution allowed us to define and use this parameter.

Additionally, we analytically showed that the PL intensity is proportional to $\eta_{nf/em}$ (see next reviewer’s remark below).

→ explanation was improved in page 20 of the revised manuscript (see **green parts** in the revised manuscript)

On one hand, Fig. (7) does not compare on the same graph this quantity with the data which are in Fig 3-4. There is only a trend agreement. On the other hand, the introduction of this quantity is not motivated. If it is expected that it varies like the signal, it should be derived starting from Eq. (1).

The same remark applies to δ' . It should be specified that max and min are defined when η varies as a function of angle (presumably). It is unclear if the agreement between δ and δ' is accidental or whether it is expected on the basis of some physical arguments. Without a physical argument predicting this behaviour, in my opinion, there are not enough evidences to draw any useful conclusion at this stage.

We thank reviewer 1 for this very important remark. The definition of the overlap integral has been slightly modified (see new Eq. 4, page 20) and the expression of the PL intensity (IPL) has been derived from Eq. 1, allowing us to find out an analytical link between the overlap integral, $\eta_{nf/em}$ and the PL intensity, IPL. It actually turns out that $\eta_{nf/em}$ is proportional to IPL, making δ_{PL} (Eq. 3) = $\delta_{nf/em}$ (defined in Eq. 7), and thus allowing for direct comparison between Fig 7 and Figures 2-4.

→ see added **green parts**, pages 20-23, in the revised manuscript

Reviewer #2 (Remarks to the Author):

The manuscript presents the preparation and characterization of hybrid light emitters based on plasmonic particles and quantum dots. The fabrication method is based on plasmon-enhanced two-photon polymerization and enables the authors to position the quantum dots at different positions around the metal particles. For optical characterization, two quantities are defined: the spatial overlap integral between emitters and the plasmonic field, and an excitation polarization contrast.

Positioning emitters near plasmonic particles is not novel, as the authors also acknowledge in the introduction. Several methods were developed over the years: chemical linking, AFM-based positioning, lithography methods, or DNA assembly (not cited by the authors). Part of the authors developed over the last decade methods based on photopolymerization. Compared to previous demonstrations based on photopolymerization, I am afraid that the work could be considered incremental. The possible novelty then focuses mostly on analyzing the spatial overlap integral for single nanoparticles, but still it does not provide new knowledge gain. There are no significant technical issues, but I believe that the article will not be influential enough in the (sub-)field to warrant publication in Nature Communications.

We do not share this point of view, for three reasons:

- 1) The different methods, evoked by reviewer 2, for positioning the active medium in the vicinity of metal nanostructures, would not have permitted this study in the sense that they do not permit to control the degree of symmetry of the active layer for different geometries of metal nanoparticles, as reported in our article. This is due to the fact

that our approach presents an important feature: the intrinsic plasmonic modes of the metal nanoparticles are used for positioning the active medium at strategic sites

- 2) Our paper does not aim at demonstrating plasmonic-based photo-polymerization that has indeed been already reported in previous publications. This method is “only” a powerful tool we used for carrying out our study.
- 3) As admitted by reviewer 2, the novelty lies on the study of the spatial overlap between the active medium and excitation near-field for single hybrid particle. In order to stress this point and to avoid any ambiguity about the main message of the paper, we modified some sentences in the abstract, introduction and conclusion pages (see highlighted **red parts**). This study has never been proposed because the current methods for positioning the nano-emitters probably do not allow to envisage it. The study is new and unique at this spatial scale. It allowed one to introduce and use three new parameters: i) ρ , the probability of presence of nano-emitters as a function of x,y,z (the Descartes coordinates) or r,ϕ (the polar coordinates), e.g. nano-emitters are present within $(r \in [20 \text{ nm}-25 \text{ nm}]) \cap (\phi \in [0^\circ-360^\circ])$, and associated nanoscale angular filling factor β that quantifies the way the space surrounding the metal nanoparticle is occupied by the nanoemitters, ii) the spatial overlap integral $\eta_{nf/em}$ between exciting near-field field and the nanoemitters, iii) the resulting polarization contrast of the photoluminescence, resulting in a new generation of polarization-driven hybrid nanoemitters.

The high level of control of the nanophotopolymerization process allowed one to define and discuss these parameters.

Furthermore, thanks to reviewer 1, we demonstrated that the photoluminescence intensity is proportional to the $\eta_{nf/em}$, which we consider as to be a very important result.

Physics behind this parameter is rich, as illustrated by, e.g., the discussion in page 14, lines 4-17.

We stress that the optimization of these parameters is likely to open new avenues in the near future, especially concerning polarization-driven tunable nano-emitters including nanolasers and single-photon emitters. In particular, regarding nanolasers, it is reasonable to assume that the effective pumping is directly linked to the overlap integral $\eta_{nf/em}$. Polarization could thus command both the pumping and the color in the case of anisotropic nanoscale distribution of colors.

→ In order to highlight these novelties, we improved the manuscript to make the message clearer (see added **red** parts throughout the manuscript)

→ Finally, in order to consider the reviewer’s skepticism, we made an important decision: we decided to add some very recent results that were initially destined to other publications: concentration of QDs within the photosensitive formulation was decreased to obtain nanocube-based hybrid nanosources containing single QDs. Preliminary demonstration of a polarization-sensitive single-photon switch has been added to the manuscript (see underlined parts). The title of the manuscript has been consequently modified.

→ As suggested by reviewer 2, we also added three references on the use of DNA assembly for positioning nano-emitters in the vicinity plasmonic nanostructures (ref.

49, 50, 51 in the revised manuscript). We apologize for this omission in the initial manuscript but notice that this interesting approach does not allow the full control of the position of the nano-emitters in the sense that only gaps between coupled metal nanoparticles can get functionalized with nano-emitters.

The manuscript might benefit from attention to the following points:

We thank reviewer2 for highlighting these important following points.

- Equation 1 seems to neglect Stokes shift (wavelength dependence of excitation and emission).

Equation 1 has been modified and some comments have been added
→ see pink parts, pages 4 and 5 of the revised manuscript

- Please substantiate the claim that “the HPNs are stable over time. In particular, they do not suffer from photobleaching” with data. How long were the HPNs exposed to light to reach this conclusion? How? Please quantify it.

We admit that this conclusion was a bit premature. The nano-emitters just looked stable for a few hours during our experiments. To take into account this relevant remark, we measured emission from four different nano-cube based hybrid nano-emitters that have been shined for 30 minutes every day at $\lambda=405$ during a period of many days, up to 25 days. Measurements were made regularly and randomly during this period. It turns out that the emission actually drops over time, as a probable consequence of either photo bleaching or chemical degradation due to, e.g., oxidation of QDs, but remains pretty stable for about 5 days: 10%-50% drop, depending on the considered nano-objects and the incident intensity. This study has been added as a supplementary information (section 6, Fig. S6). The origin of this drop and related processes will have to be studied in the near-future.

→ see new khaki parts in page 12 of the revised manuscript and supplementary information file, section 6

- Can the authors at least estimate the number of quantum dots attached to each nanocube?

This is an important issue. We have been asking ourselves the same question since the beginning of the project. Quantifying the number of involved QDs is of importance. This is why we have been struggling to perform the plasmon-based photo polymerization experiments on TEM grids, in order to get the possibility to directly observe and count the QDs. Unfortunately, so far, we did not succeed in doing so, for several reasons, including the fragility of the substrate. However, we succeeded in polymerizing a drop of QDs-containing formulation on a grid, permitting TEM observation of QDs at the edges of the solidified drop where polymer thickness is very

low. As a result, an assessment of the number of involved QDs was made: a few tens of QDs / polymer lobe for the biggest lobe has been assessed.

→ see added **blue parts** in both the revised manuscript (page 11) and updated supplementary information file (section 2)

- Figure 4b: please describe better the meaning of this image composition, with more accuracy, as it is open to interpretation. Is it precisely the same GNC that is covered with polymer in Figure 4a? It is strange to merge two SEM images instead of highlighting the profile of the GNC on the final nanostructure.

Fig. 4a is SEM image of the final nanostructure. In figure 4b, 4a was superimposed with the SEM image of the same gold nanocube, taken before the photopolymerization procedure. This allows us to get a clear top view of the integrated polymer. In order to take into account this remark, AFM profiles have been added in supplementary information

→ see added **gray parts** in both manuscript (page 11 + captions of Figures 2 and 3) and supplementary information (section 4).

Ps: There is an incomplete sentence: “Compared to the nanocube case, the dipolar near field distribution yields of 2PP process with reduced control over”.

The sentence has been completed (same remark from reviewer 1).

REVIEWERS' COMMENTS:

Reviewer #1 (Remarks to the Author):

I have read the revised manuscript. The modifications suggested have been implemented. The introduction of the overlap figure of merit is convincing.

Finally the authors have added new results. They have shown that this technique allows to fabricate a single photon source with a Purcell factor of 23. In addition, it can be controlled using far-field polarization.

The second referee has indicated that this work does not introduce a novel technique to deal with the issue of positioning emitters close to nanostructures. In my view, the novelty is to develop a technique that ensures a good overlap between the location of the emitters and the maxima of the electric field. This is well characterized by the figure of merit introduced by the authors. Demonstrating that it can be used to fabricate a single photon emitter is a significant achievement.

As the near-field distribution depends strongly on polarization, the result is a system that can be addressed selectively by controlling the exciting polarization in the far field.

I recommend publication.

Reviewer #2 (Remarks to the Author):

The authors have addressed my technical concerns. My novelty concerns for the contents of the first version remain mostly unchanged: other methods were able to position quantum emitters with the same or higher accuracy than the present results. In my opinion, if the spatial overlap between excitation and active medium was not studied in this detail in the past it was because of lack of interest (incremental knowledge, no surprises in the physics, just painful to carry out the study with statistical significance).

The authors have added a new section that is certainly interesting about single a quantum dot structure showing polarization switching. It is a valuable although preliminary addition.

At this point, and given the subjective nature of assessments on the degree of novelty and significance, I believe acceptance or rejection should be an editorial decision based on the level of novelty expected for a Nature Communications, taking into account the added value of the new data with a single quantum dot but its incomplete character showing a single structure.

ps: note that Figure 8c says "Times (s)" on the vertical axis label.

Manuscript NCOMMS-19-33496B

Response to the referee's comments

We thank very much the reviewers for the second review of the manuscript entitled "Hybrid plasmonic nano-emitters: control of the spatial overlap between the local excitation field and the active medium down to a single quantum emitter" (new title is "Hybrid plasmonic nano-emitters with controlled single quantum emitter positioning on the local excitation field").

These new reviews are commented below:

Reviewer #1 (Remarks to the Author):

I have read the revised manuscript. The modifications suggested have been implemented. The introduction of the overlap figure of merit is convincing. Finally, the authors have added new results. They have shown that this technique allows to fabricate a single photon source with a Purcell factor of 23. In addition, it can be controlled using far-field polarization. The second referee has indicated that this work does not introduce a novel technique to deal with the issue of positioning emitters close to nanostructures. In my view, the novelty is to develop a technique that ensures a good overlap between the location of the emitters and the maxima of the electric field. This is well characterized by the figure of merit introduced by the authors. Demonstrating that it can be used to fabricate a single photon emitter is a significant achievement. As the near-field distribution depends strongly on polarization, the result is a system that can be addressed selectively by controlling the exciting polarization in the far field. I recommend publication.

We are glad to see that referee #1 thinks that our work is definitely of interest and deserves publication in Nature Communications. He/she admirably summarized the main assets of our approach. We thank him/her warmly for both support and relevant suggestions.

Reviewer #2 (Remarks to the Author):

The authors have addressed my technical concerns. My novelty concerns for the contents of the first version remain mostly unchanged: other methods were able to position quantum emitters with the same or higher accuracy than the present results. In my opinion, if the spatial overlap between excitation and active medium was not studied in this detail in the past it was because of lack of interest (incremental knowledge, no surprises in the physics, just painful to carry out the study with statistical significance).

The authors have added a new section that is certainly interesting about single a quantum dot structure showing polarization switching. It is a valuable although preliminary addition.

*At this point, and given the subjective nature of assessments on the degree of novelty and significance, I believe acceptance or rejection should be an editorial decision based on the level of novelty expected for a Nature Communications, taking into account the added value of the new data with a single quantum dot but its incomplete character showing a single structure.
ps: note that Figure 8c says "Times (s)" on the vertical axis label.*

Thanks for this second review. Thanks for the remark about Figure 8c ('s' has been removed from 'Times')

We are sorry to see that referee #2 is still not convinced that our work is original enough, although he/she admits that the technical concerns have been addressed and that the added new section is valuable.

We respect the referee #2' point of view but we don't share it at all. In particular, we do not agree with the two following statements:

- 1) *other methods were able to position quantum emitters with the same or higher accuracy than the present results*

We do not think that other methods could have been able to position quantum emitters with so many degrees of symmetry:

This is due to the fact that our approach has a major asset: it uses the intrinsic plasmonic field to position the quantum emitters through local plasmonic photopolymerization. If referee #2 wants to provide the reference of a published work which is susceptible to oppose our point of view, we would be happy to consider this new element (referee #2 can provide this reference via the Nature Communications editorial office)

2) In my opinion, if the spatial overlap between excitation and active medium was not studied in this detail in the past it was because of lack of interest (incremental knowledge, no surprises in the physics, just painful to carry out the study with statistical significance).

We don't agree. Controlling this overlap is actually crucial as it is in classical optoelectronics. As pointed out in the conclusion of the manuscript, this new class of anisotropic plasmonic nano-emitters opens up the avenue for polarization-driven tunable nano-emitters including nanolasers and single-photon emitters. In particular, regarding nanolasers, it is reasonable to assume that the effective pumping intensity is directly dependent on the overlap integral discussed in the article. Polarization could thus command both the pumping process and the associated color in the case of anisotropic nanoscale distribution of differently colored nano-emitters.

The development of these anisotropic hybrid nano-emitters has not been "painful". However, it has been difficult because a lot of complex parameters had to be controlled, including the photochemical parameters for nano-emitter fabrication. The control of these complex parameters has been leading to many new achievements of observations. The reported study is thus valuable and worth being shared. We did our best to share the technical issues that will hopefully allow other groups to reproduce the experiments. We will be happy to collaborate on the use of these hybrid nano-emitters in the future.

Anyway, we would like to thank referee #2 again for reviewing our manuscript twice and for raising important technical issues, despite the significant difference of scientific opinion.

→The second reviews have resulted in some added sentences in the revised manuscript (page 27, lines 14-19)

Renaud Bachelot, on behalf of the authors